# HypRQ-VAE: Long-Tail-Aware Item Indexing for Generative Recommender Systems

## Abstract

Sequential recommender systems model user behavior as item-ID sequences, while recent generative methods cast recommendation as a language modeling task using large language models (LLMs). While this paradigm incorporates rich textual semantics, it creates a fundamental mismatch: LLMs operate on text tokens, whereas recommender systems depend on discrete item indices. This misalignment often leads to hallucinations in generative recommendations. Existing methods attempt to bridge this gap by learning item vocabularies in Euclidean space, but they struggle to model the inherent long-tail distribution of real-world catalogs, where a small number of head items dominate and a vast number of tail items reflect users' niche preferences. To address this issue, we introduce Hyperbolic Residual-Quantized Variational AutoEncoder (HypRQ-VAE), the first framework to learn item indexing in hyperbolic space. HYPRQ-VAE leverages the unique properties of hyperbolic geometry, whose exponential volume expansion naturally accommodates the power-law structure of user-item interactions. This allows the model to encode rich textual semantics while preserving the representational fidelity of sparse, long-tail items. Experiments on three benchmark datasets show that HYPRQ-VAE significantly improves the performance of recommendation, particularly in recommending tail items. Our analysis attributes these gains to the superior capacity of hyperbolic space to model item hierarchies and sparsity in generative recommendation. Our code and data are available at: `https://anonymous.4open.science/r/HypRQ-VAE-6C5B`.

## 1 Introduction

In today's data-saturated digital world, recommender systems (RS) play a crucial role in alleviating information overload by delivering personalized and relevant content, which streamlines users' decision-making and boosts service efficiency. Since user preferences dynamically evolve over time, sequential recommendation has attracted significant attention for its ability to capture the temporal patterns in user behavior. Most state-of-the-art models are built on sequentially created interaction logs where each event is encoded by an item ID (Kang & McAuley, 2018; Tang & Wang, 2018). To model these sequences, researchers have explored a wide range

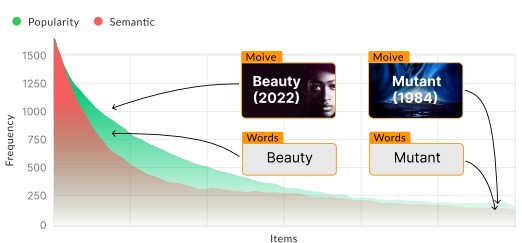

Figure 1: An illustrative figure of long-tail distribution in item popularity and semantic tokens.

of deep learning architectures, such as GNNs (He et al., 2020) and Transformers (Sun et al., 2019), and enriched the basic item-ID signal with auxiliary content such as titles, descriptions, and categories. More recently, pre-trained language models (PLMs) have been introduced to better exploit the textual semantics in item metadata, further improving recommendation performance (Liu et al., 2023; Chen et al., 2023; Wei et al., 2024).

The remarkable success of Large Language Models (LLMs) across diverse language tasks (Achiam et al., 2023; Touvron et al., 2023; Radford et al., 2019) has inspired their application to RS by framing recommendation as a generation problem (Geng et al., 2022; Bao et al., 2023). However, a

fundamental challenge persists: the semantic misalignment between the language representations that LLMs excel at processing and the collaborative filtering signals foundational to RS. Traditional recommenders model user behavior as sequences of discrete item IDs, while LLMs operate on textual tokens. This vocabulary mismatch hinders the effective use of LLM for generating accurate recommendations. Although items often come with textual information through titles or descriptions, these texts often comprise tens to thousands of tokens, making it exceedingly difficult for LLMs to generate precise, real-world item recommendations without hallucination (Sriramanan et al., 2024). Conversely, using raw item IDs (e.g., '0123456789') leads to an unmanageably large, semantically meaningless vocabulary (e.g., <01><23><45>...) Geng et al. (2022), which fails to capture relationships among items. To bridge this gap, recent works have proposed specialized item-indexing mechanisms that construct item vocabularies and train LLMs to generate target items directly (Rajput et al., 2023; Zheng et al., 2024). A notable example is the Residual-Quantized Variational AutoEncoder (RQ-VAE) Zeghidour et al. (2021), a multi-level vector quantizer that recursively quantizes the residual vectors, from coarse to fine, to generate a set of codewords, producing item indices that capture the hierarchical structure of items. However, these methods typically operate in Euclidean space, which struggles with the long-tail distribution prevalent in real-world recommendation data.

Large-scale recommendation datasets commonly exhibit a pronounced long-tailed distribution (Liu et al., 2024): a small fraction of popular "head" items dominate user interactions, while a vast number of "tail" items reflect niche interests and emerging trends. Both of them are essential for a well-rounded recommender system. Meanwhile, item semantics also follow the long-tailed distribution, as illustrated in Figure 1. With these two distributions sharing implicit correlations, popular items typically exhibit more generic semantic content, while rare items display greater semantic distinctiveness. However, traditional Euclidean methods disproportionately emphasize head items while neglecting tail items (Yang et al., 2022). To better model this inherent imbalance, researchers have turned to hyperbolic space, whose exponentially expanding volume naturally accommodates the power-law structure of user-item interactions. Although hyperbolic recommendation models have shown promising empirical results (Sun et al., 2021; Yang et al., 2022), two key questions remain unsolved: *(1) How effective are hyperbolic models at generating item IDs?* and *(2) Do hyperbolic models outperform Euclidean approaches, especially for tail-items recommendation?*

To answer the above questions, we introduce Hyperbolic RQ-VAE (HYPRQ-VAE), a novel framework that generates semantic IDs for items in hyperbolic space. Our goal is to leverage hyperbolic geometry's exponentially increased capacity and its inherent ability for modeling hierarchical structures to provide greater representational flexibility, particularly for rare semantic concepts. We conduct a comparative analysis between hyperbolic and Euclidean models, partitioning items into the top 20% "head" (H20) and bottom 80% "tail" (T80) based on the Pareto Principle [1]. Our experiments reveal that hyperbolic models allocate substantially more representational capacity to tail items than Euclidean models (see Section 4.3). We argue that hyperbolic geometry, with its high representational capacity and its natural fit for hierarchical data, can yield richer and more discriminative representations for tail items than Euclidean embeddings. While prior research has incorporated various signals for item indexing, to the best of our knowledge, this is the first work to incorporate hyperbolic representation into the item indexing process for generative recommendation. Our main contributions are summarized as follows:

- **Novel Perspective.** We are the first to explore the problem of item indexing in the hyperbolic space for generative recommendation tasks.

- **New Framework.** We introduce HYPRQ-VAE, an indexing framework that integrates rich item semantics with a hyperbolic geometry naturally suited to the long-tailed distribution characteristic of recommender systems.

- **Empirical Studies.** We conduct extensive experiments on three widely adopted datasets (Instruments, Arts from the Amazon, and MovieLens) with standard sequential models and competing indexing strategies, demonstrating the effectiveness of our approach, especially on tail items. Our in-depth analysis highlights how hyperbolic representations contribute to these improvements.

---

[1] https://en.wikipedia.org/wiki/Pareto_principle

## 2 PRELIMINARIES

**Riemannian Geometry.** A $d$-dimensional Riemannian manifold $\mathcal{M}$, denoted as $(\mathcal{M}, \mathfrak{g})$, is a topological space equipped with a metric tensor $\mathfrak{g}$ (Do Carmo & Flaherty Francis, 1992). At any point $\mathbf{x}$ in $\mathcal{M}$, the manifold can be locally approximated by a local linear space called the tangent space $\mathcal{T}_{\mathbf{x}}\mathcal{M}$, which is isometric to $\mathbb{R}^d$. The geodesic represents the shortest path between two points on the manifold (Alexander & Alexander, 1981), and its length defines the induced distance $d_{\mathcal{M}}$. Notably, in hyperbolic space, the hyperbolic distance to the origin (HDO) serves as the induced hyperbolic norm. The exponential map $\exp_{\mathbf{x}}$ projects tangent vectors onto the manifold, and its reverse function, the logarithmic map $\log_{\mathbf{x}}$, returns points to the tangent space. Furthermore, parallel transport $P_{\mathbf{x} \to \mathbf{y}}$ enables the transportation of geometric data from $\mathbf{x}$ to $\mathbf{y}$ along the unique geodesics while preserving the metric tensor $\mathfrak{g}$.

**Hyperbolic Space.** Hyperbolic space refers to a Riemannian manifold with a constant negative curvature, and its coordinates can be represented using various isometric models (Cannon et al., 1997; Ramsay & Richtmyer, 2013). In the domain of machine learning and deep learning, the Poincaré ball model has emerged as a particularly valuable framework, offering unique advantages for embedding hierarchical structures and complex relational data in a bounded continuous space (Nickel & Kiela, 2017; Tifrea et al., 2018; Balazevic et al., 2019; Yang et al., 2023). A $d$-dimensional Poincaré ball model with a constant negative curvature $\kappa (\kappa < 0)$ is defined as a Riemannian manifold

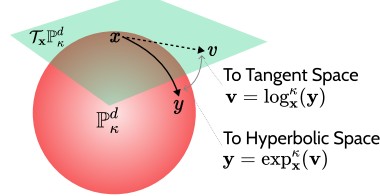

Figure 2: Illustration of the Poincaré ball model and its associated exponential and logarithmic maps.

fold $(\mathbb{P}_{\kappa}^d, \mathfrak{g}^{\kappa})$, where $\mathbb{P}_{\kappa}^d = \{ x \in \mathbb{R}^d | \|\mathbf{x}\|^2 < -\frac{1}{\kappa} \}$ and its metric tensor $\mathfrak{g}^{\kappa} = (\lambda_{\mathbf{x}}^{\kappa})^2 \mathbf{I}_d$ and $\mathbf{I}_d$ is $d$-dimensional identity matrix. Here, $\mathbb{P}_{\kappa}^d$ is an $d$-dimensional open ball of radius $(-\kappa)^{-\frac{1}{2}}$, and $\lambda_{\mathbf{x}}^{\kappa} = 2(1 + \kappa \|\mathbf{x}\|^2)^{-1}$ is the conformal factor. This induces the inner product $\langle \mathbf{p}, \mathbf{q} \rangle_{\mathbf{x}}^{\kappa} = (\lambda_{\mathbf{x}}^{\kappa})^2 \langle \mathbf{p}, \mathbf{q} \rangle$ and norm $\|\mathbf{p}\|_{\mathbf{x}}^{\kappa} = \lambda_{\mathbf{x}}^{\kappa} \|\mathbf{p}\|$ for $\mathbf{p}, \mathbf{q} \in \mathcal{T}_{\mathbf{x}}\mathbb{P}_{\kappa}^d$. The exponential and logarithmic maps, denoted $\exp_{\mathbf{x}}^{\kappa}$ and $\log_{\mathbf{x}}^{\kappa}$, are defined below.

**Exponential and Logarithmic Map.** This framework defines several operations, including Möbius addition $\oplus_{\kappa}$, distance between points on the manifold $d_{\kappa}(\mathbf{x}, \mathbf{y})$, exponential and logarithmic maps. Given their relevance to our work, we present the definitions of the exponential and logarithmic maps below. These operations are defined within the Riemannian manifold framework, specifically considering the Poincaré ball model $\mathbb{P}_{\kappa}^d$:

$$\mathbf{p} \oplus_{\kappa} \mathbf{q} = \frac{(1 - 2\kappa\langle \mathbf{p}, \mathbf{q} \rangle - \kappa\|\mathbf{q}\|^2)\mathbf{p} + (1 + \kappa\|\mathbf{p}\|^2)\mathbf{q}}{1 - 2\kappa\langle \mathbf{p}, \mathbf{q} \rangle + \kappa^2\|\mathbf{p}\|^2\|\mathbf{q}\|^2},$$

$$\exp_{\mathbf{x}}^{\kappa}(\mathbf{v}) = x \oplus_{\kappa} (\tanh(\sqrt{-\kappa}\frac{\lambda_{\mathbf{x}}^{\kappa}\|\mathbf{v}\|}{2})\frac{\mathbf{v}}{\sqrt{-\kappa}\|\mathbf{v}\|}), \tag{1}$$

$$\log_{\mathbf{x}}^{\kappa}(\mathbf{v}) = \frac{2}{\sqrt{-\kappa}\lambda_{\mathbf{x}}^{\kappa}}(\tanh^{-1}(\sqrt{-\kappa}\| - \mathbf{x} \oplus_{\kappa} \mathbf{v}\|)\frac{-\mathbf{x} \oplus_{\kappa} \mathbf{v}}{\| - \mathbf{x} \oplus_{\kappa} \mathbf{v}\|}),$$

The induced Möbius substraction $\ominus_{\kappa}$ is given by $\mathbf{p} \ominus_{\kappa} \mathbf{q} = \mathbf{p} \oplus_{\kappa} (-\mathbf{q})$. And the exponential and logarithmic maps enable transitions between the tangent space and the hyperbolic space (as shown in Figure 2).

## 3 METHOD

### 3.1 OVERVIEW OF THE APPROACH

In Section 1, we discussed a key limitation of using LLMs for recommendation: the mismatch between the textual semantics captured by LLMs and the collaborative semantics essential for effective recommendation. To address this fundamental limitation, we propose a two-stage strategy that enhances semantic integration.

**Generation of Semantic IDs.** We use hyperbolic RQ-VAE to compress the textual embedding of each item into several learned discrete IDs. Specifically, we project the representation of items from

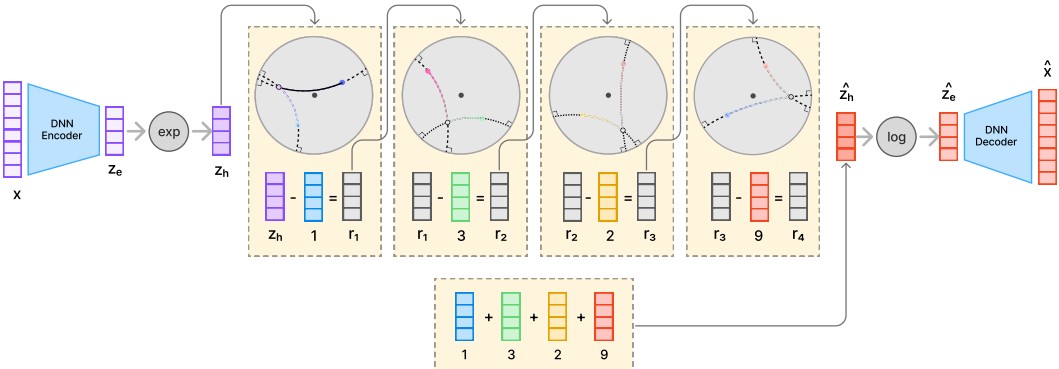

Figure 3: HYPRQ-VAE: The figure depicts the encoding process for an item. An item embedding $\mathbf{x}$ is passed through a DNN encoder to obtain a latent vector $z_e$. This vector is then projected into hyperbolic space via an exponential map to procude $z_h$, which serves as the initial residual $r_0$ (purple bar). This residual is then processed sequentially through multiple quantization levels. At each level $i$, the quantizer identifies the nearest code vector $e_{c_i}^{l_i}$ from its codebook and computes the next residual $r_{i+1} = r_i \ominus_\kappa e_{c_i}^{l_i}$ using Möbius subtraction. For instance, at the first level, the code vector $e_{c_1}^{l_1}$ (blue bar) is selected, and at the second level, $e_{c_2}^{l_2}$ (green bar) is chosen. The final semantic code is the sequence of indices of these selected vectors, exemplified by $(1, 3, 2, 9)$.

tangent space into hyperbolic space via exponential and logarithmic maps, enabling the learned item indices to better capture similarities between the textual semantics of items while providing unique indexing representations for specific items. Furthermore, hyperbolic geometry's exponential volume growth and natural support for hierarchical structure further boost representational capacity, especially for long-tail items, which tend to receive less attention in conventional Euclidean models.

**Generative Recommender with Semantic IDs.** To bridge the semantic gap, we introduce a series of prompt instructions aligning textual semantics and collaborative semantics. By fine-tuning on sequences of semantic IDs, we integrate collaborative knowledge into the LLM's generative process, enabling it to make more effective recommendations.

## 3.2 GENERATION OF SEMANTIC IDS

A key challenge in adapting LLMs for recommendation is encoding items into a compact set of discrete tokens. We address this by generating a Semantic ID for each item in the hyperbolic space, a structured sequence of learned indices derived from its textual features. Our method is built upon RQ-VAE. We reformulate the quantization in hyperbolic space, whose geometric properties are better suited for modeling the hierarchical and long-tail nature of item data.

Our model first uses an encoder to map an item's embedding $\mathbf{x}$ into a Euclidean latent representation $z_e \in \mathbb{R}^d$. This vector is then projected onto the poincaŕe ball $\mathbb{P}_\kappa^d$ using the exponential map $\exp_0^\kappa$, yielding the projected latent representation $z_h \in \mathbb{P}_\kappa^d$. This vector $z_h$ serves as the initial residual, $r_0$, for the quantization process. At each quantization level $l$, we have a codebook $\mathcal{C}^l = \{e_k^l\}_{k=1}^K$, where $K$ is the dimension of the codebook, and each code vector $e_k^l$ is a learnable cluster center. The residual quantization process is performed recursively within the hyperbolic space. At each level, we identify the index $c_i$ of the closest code vector and compute the residual for the next level, $r_{l+1}$, using Möbius subtraction:

$$c_i = \arg\min_k ||r_i \ominus_\kappa e_k^i||_2^2, \qquad r_{i+1} = r_i \ominus_\kappa e_{c_i}^i. \tag{2}$$

The final quantized representation $\hat{z_h}$ is constructed by aggregating the selected code vectors using Möbius addition: $\hat{z_h} = e_{c_1}^1 \oplus_\kappa \cdots \oplus_\kappa e_{c_L}^L$. This hyperbolic vector is then projected back to Euclidean space with the logarithmic map $\log_0^\kappa$ before being passed to the decoder to reconstruct the original embedding $\hat{\mathbf{x}}$.

Figure 4: An overview of our Transformer decoder-based architecture for generative recommendation using semantic IDs. The figure illustrates the Supervised Fine-Tuning (SFT) stage, where a causal attention mask ensures auto-regressive learning by forcing the model to predict the next token based only on preceding tokens.

The model is trained by minimizing the loss function below, $\mathcal{L}_{\text{RQ-VAE}}$:

$$\mathcal{L}_{\text{RECON}} = ||\mathbf{x} - \hat{\mathbf{x}}||_2^2,$$

$$\mathcal{L}_{\text{RQ}} = \sum_{i=1}^{L} ||\text{sg}[r_i] - e_{c_i}^i||_2^2 + \beta ||r_i - \text{sg}[e_{c_i}^i]||_2^2, \quad (3)$$

$$\mathcal{L}_{\text{RQ-VAE}} = \mathcal{L}_{\text{RECON}} + \gamma \mathcal{L}_{\text{RQ}},$$

where $\mathcal{L}_{\text{RECON}}$ is the reconstruction loss and $\mathcal{L}_{\text{RQ}}$ is the quantization loss that optimizes the codebook embeddings. The term $\text{sg}[*]$ denotes the stop-gradient operator, while $\beta$ and $\gamma$ are weighting coefficients. The sequence of indices $(c_1, \cdots, c_L)$ generated during quantization forms the core of the Semantic ID. As shown in Figure 3, an index sequence like $(1, 3, 2, 9)$ is formatted into the semantic ID of this item as <a_1><b_3><c_2><d_9>.

Unlike conventional vector quantization, RQ provides significantly larger representational capacity with a smaller codebook. Moreover, its coarse-to-fine quantization naturally produces a hierarchical item index, which aligns naturally with autoregressive generation. When integrating RQ into hyperbolic space, these benefits are amplified: the exponential expansion of hyperbolic space allows more expressive and hierarchically structured embedding, enhancing representation and understanding of the diverse items, including both head and long-tail items.

**Handling Collisions.** Compressing item content into discrete semantic IDs can cause collisions, where different items are mapped to the same tokens. Prior works mitigated this issue by appending auxiliary identifiers for conflicting items (Rajput et al., 2023; Hua et al., 2023), which can introduce semantically unrelated information. Specifically, LC-Rec (Zheng et al., 2024) employs a uniform distribution constraint in the final layer to minimize item indexing conflicts, while LETTER (Wang et al., 2024) introduces regularizers to enhance code assignment diversity. However, these methods do not completely resolve collisions, especially when the number of colliding items exceeds the capacity of the last-level codebook or when semantically similar items are inherently difficult to distinguish.

To address this challenge, we utilize a systematic token reassignment methodology based on codebook proximity metrics. Given a set of $M$ colliding items, we construct a distance tensor $\mathbf{D} \in \mathbb{R}^{M \times L \times K}$, where each element $d_k^i \in \mathbf{D}$ represents the hyperbolic distance between the residual vector $r_i$ and code vector $e_k^i$ across all codebook levels. After sorting these distances along the code dimension to generate an index $\mathbf{I}$. We implement a cascading assignment strategy starting from the last level. Each item is initially assigned its nearest available token. In case of conflicts, the item with the smallest distance to that token is assigned its index, while the other items consider the next closest available token. If the last level cannot accommodate all items with unique tokens, we proceed to the second-to-last level and repeat this process, continuing iteratively up the levels until every item is assigned a unique identification.

Table 1: Comparison of methods across three datasets. H and N denote Hit Rate and NDCG, respectively. Bold numbers indicate the best results, while numbers with an underline indicate the second-best. The final row depicts the % improvement with our method over the strongest baseline.

| Method | MovieLens | | | | Instruments | | | | Arts | | | |
|---|---|---|---|---|---|---|---|---|---|---|---|---|
| | H@5 | H@10 | N@5 | N@10 | H@5 | H@10 | N@5 | N@10 | H@5 | H@10 | N@5 | N@10 |
| MF | 0.0343 | 0.0609 | 0.0220 | 0.0305 | 0.0444 | 0.0551 | 0.0364 | 0.0399 | 0.0202 | 0.0262 | 0.0162 | 0.0181 |
| Caser | 0.0422 | 0.0651 | 0.0276 | 0.0349 | 0.0525 | 0.0706 | 0.0479 | 0.0541 | 0.0267 | 0.0394 | 0.0191 | 0.0232 |
| SASRec | 0.0518 | 0.0936 | 0.0274 | 0.0435 | 0.0596 | 0.0717 | 0.0336 | 0.0405 | 0.0381 | 0.0519 | 0.0212 | 0.0269 |
| P5-TID | 0.0228 | 0.0281 | 0.0157 | 0.0175 | 0.0002 | 0.0002 | 0.0001 | 0.0001 | 0.0006 | 0.0006 | 0.0004 | 0.0005 |
| P5-CID | 0.0612 | 0.0920 | 0.0398 | 0.0503 | 0.0537 | 0.0626 | 0.0470 | 0.0498 | 0.0400 | 0.0498 | 0.0324 | 0.0355 |
| TIGER | 0.0565 | 0.0856 | 0.0386 | 0.0480 | 0.0608 | 0.0716 | 0.0529 | 0.0563 | 0.0412 | 0.0506 | 0.0337 | 0.0367 |
| LC-Rec | 0.0553 | 0.0871 | 0.0355 | 0.0456 | 0.0620 | 0.0780 | 0.0530 | 0.0570 | 0.0413 | 0.0505 | 0.0340 | 0.0369 |
| LETTER | 0.0649 | 0.0964 | 0.0410 | 0.0535 | 0.0600 | 0.0730 | 0.0520 | 0.0550 | 0.0383 | 0.0495 | 0.0317 | 0.0353 |
| Ours | **0.0652** | **0.1010** | **0.0423** | **0.0539** | **0.0690** | **0.0830** | **0.0600** | **0.0650** | **0.0425** | **0.0528** | **0.0348** | **0.0381** |
| | + 0.5% | + 4.8% | + 3.2% | + 0.7% | + 11.3% | + 6.4% | + 13.2% | + 14.0% | + 2.9% | + 4.3% | + 2.4% | + 3.3% |

## 3.3 GENERATIVE RECOMMENDER WITH SEMANTIC IDS

We formulate next-item prediction as a sequence-to-sequence generation task over semantic IDs. As illustrated in Figure 4, a HypRQ-VAE is first trained on the textual content of items in the user's Interaction History to encode each item into a discrete, $l$-length Semantic ID $(c_{i,1}, \cdots, c_{i,l})$. A user's history is then represented as a chronologically ordered sequence of these IDs. This sequence is flattened into a single token stream: $(c_{1,1}, \cdots, c_{1,l}, c_{2,1}, \cdots, c_{2,l}, \cdots, c_{n,1}, \cdots, c_{n,l})$, and subsequently embedded within a prompt template to form the full input sequence (see Appendix C for an example).

The model is trained in an autoregressive manner to generate the ID of the next item, $(c_{n+1,1}, \cdots, c_{n+1,l})$. We optimize the model by minimizing the negative log-likelihood of the target sequence:

$$\mathcal{L}_\theta = -\sum_{j=1}^{|\mathbf{Y}|} \log P_\theta(\mathbf{Y}_j|\mathbf{Y}_{<j}, \mathbf{X}), \tag{4}$$

where $\theta$ represents the model parameters, $\mathbf{X}$ is the input prompt containing the historical item IDs, $\mathbf{Y}$ is the target item's ID sequence, and $\mathbf{Y}_{<j}$ are all tokens in the target sequence before the $j$-th token.

During inference, the goal is to generate the next item that best aligns with the user's preferences. The model autoregressively produces the sequence of semantic ID tokens: $\hat{\mathbf{Y}}_t = \arg\max_{v \in \mathcal{V}} P_\theta(v|\hat{\mathbf{Y}}_{<t}, \mathbf{X})$, where $\mathcal{V}$ is the vocabulary including all possible semantic ID tokens.

## 4 EXPERIMENTS

## 4.1 EXPERIMENTAL SETTINGS

**Datasets.** We evaluate our model on three widely adopted real-world recommendation datasets: MovieLens [2], and the Instruments and Arts from the Amazon [3] dataset (Ni et al., 2019). Detailed information and the statistics are provided in Appendix A.

**Baselines.** We compare our model, HYPRQ-VAE against a strong set of baselines (see Appendix B for a brief description): MF (Rendle et al., 2012), a matrix factorization model with BPR loss; Caser (Tang & Wang, 2018), which uses CNNs to capture sequential patterns; and SASRec (Kang & McAuley, 2018), a self-attention-based model. P5 (Hua et al., 2023), with two different item identifier strategies: title-based (TID) and collaborative-based (CID); TIGER (Rajput et al., 2023), which uses a Euclidean RQ-VAE for codebook-based identifiers; LC-Rec (Zheng et al., 2024) and LETTER (Wang et al., 2024), two recent models that also generate codebook-based identifiers with advanced alignment and semantic integration techniques.

---

[2] https://movielens.org/
[3] https://jmcauley.ucsd.edu/data/amazon/

Table 2: Performance comparison of Euclidean and hyperbolic methods on the three datasets. The relative improvements compared with Euclidean method is denoted by $\Delta_{\mathcal{H}}$.

| Metric | Method | MovieLens | | | Instruments | | | Arts | | |
|---|---|---|---|---|---|---|---|---|---|---|
| | | H20 | T80 | All | H20 | T80 | All | H20 | T80 | All |
| Hit@5 | TIGER | 0.0872 | 0.0103 | 0.0565 | 0.0982 | 0.0013 | 0.0608 | 0.0691 | 0.0031 | 0.0412 |
| | Ours | 0.0914 | 0.0153 | 0.0652 | 0.1099 | 0.0018 | 0.0690 | 0.0707 | 0.0036 | 0.0425 |
| | $\Delta_{\mathcal{H}}$ | + 4.82% | + 48.54% | + 15.52% | + 11.88% | + 31.95% | + 13.42% | + 2.32% | + 16.13% | + 3.29% |
| Hit@10 | TIGER | 0.1292 | 0.0203 | 0.0856 | 0.1161 | 0.0021 | 0.0716 | 0.0834 | 0.0058 | 0.0506 |
| | Ours | 0.1474 | 0.0310 | 0.1010 | 0.1305 | 0.0028 | 0.0830 | 0.0868 | 0.0068 | 0.1010 |
| | $\Delta_{\mathcal{H}}$ | + 14.09% | + 52.71% | + 17.97% | + 12.34% | + 36.23% | + 15.97% | + 4.08% | + 17.24% | + 5.28% |
| NDCG@5 | TIGER | 0.0578 | 0.0067 | 0.0386 | 0.0865 | 0.0008 | 0.0529 | 0.0570 | 0.0020 | 0.0337 |
| | Ours | 0.0599 | 0.0092 | 0.0423 | 0.0954 | 0.0010 | 0.0600 | 0.0583 | 0.0024 | 0.0348 |
| | $\Delta_{\mathcal{H}}$ | + 3.63% | + 37.31% | + 9.62% | + 10.23% | + 23.99% | + 13.50% | + 2.28% | + 20.00% | + 3.31% |
| NDCG@10 | TIGER | 0.0735 | 0.0099 | 0.0480 | 0.0923 | 0.0012 | 0.0563 | 0.0616 | 0.0029 | 0.0367 |
| | Ours | 0.0759 | 0.0142 | 0.0539 | 0.1020 | 0.0015 | 0.0650 | 0.0635 | 0.0035 | 0.0381 |
| | $\Delta_{\mathcal{H}}$ | + 3.27% | + 43.43% | + 12.18% | + 10.49% | + 22.56% | + 15.39% | + 3.08% | + 20.69% | + 3.84% |

**Implementation Details.** For all baselines, we use the default hyperparameters from their official implementations to ensure a fair comparison. For evaluation, we adopt the leave-one-out strategy and report top-K, Hit Rate (Hit@K), and NDCG@K for K=5, 10. Following standard practice, we perform a full ranking over all items. For all generative models, the beam size is set to 20 during inference. Further details are available in Appendix C.

## 4.2 OVERALL PERFORMANCE

We first compare HYPRQ-VAE against all baselines on the three datasets, and the overall results are shown in Table 1. Based on these results, we can find: (1) Among the LLM-based models with ID identifiers, P5-TID directly utilizes the title of items as the identifier; its relatively strongest performance on MovieLens suggests that longer token sequences increase the difficulty of accurate item prediction. (2) Models like P5-CID and LETTER demonstrate the benefit of incorporating collaborative signals, achieving strong performance on dense datasets like the MovieLens dataset. (3) Our proposed HYPRQ-VAE consistently outperforms all baselines in three datasets, highlighting the advantages of its underlying architecture. Its superior performance can be attributed to the exponentially growing capacity of hyperbolic space, which allocates more representational power to long-tail items than Euclidean embeddings, thus leading to significantly improved overall performance.

## 4.3 ANALYSIS OF LONG-TAIL PERFORMANCE

While hyperbolic models have shown promise, their specific advantages, especially for long-tail recommendation, remain underexplored. To address this, we investigate a critical question: *How does hyperbolic geometry specifically impact recommendation performance on head versus tail items?*

**Performance on Head vs. Tail Items.** To investigate the above question, we partition items in each dataset into head (top 20% most popular, H20) and tail (remaining 80%, T80) sets based on their popularity. We then compare the performance of HYPRQ-VAE against its Euclidean counterpart, TIGER (RQ-VAE). The results in Table 2 reveal a clear pattern: while HYPRQ-VAE improves performance on head items, the most substantial gains are consistently observed on tail items. For example, on MovieLens, HYPRQ-VAE improves Hit@10 by over +52.71% for tail items, a much larger margin than for head items (+14.09%).

Furthermore, as shown in Figure 5, we analyze the composition of the top-5 recommendation lists (More detailed results, including top-10 recommendations, can be found in Appendix D). The results show that HYPRQ-VAE recommends a significantly higher proportion of tail items than the Euclidean model, regardless of whether the ground truth item belongs to the head or tail. This confirms that our model does not just improve metrics but actively promotes item diversity.

**Quantifying Representation Quality.** We hypothesize that these performance gains stem from the ability of hyperbolic space to better represent sparse, long-tail items during quantization. In Euclidean space, distant and sparse tail items are often treated as outliers, leading to poor representations during quantization. In contrast, Hyperbolic space's exponential expansion and tree-like structure naturally

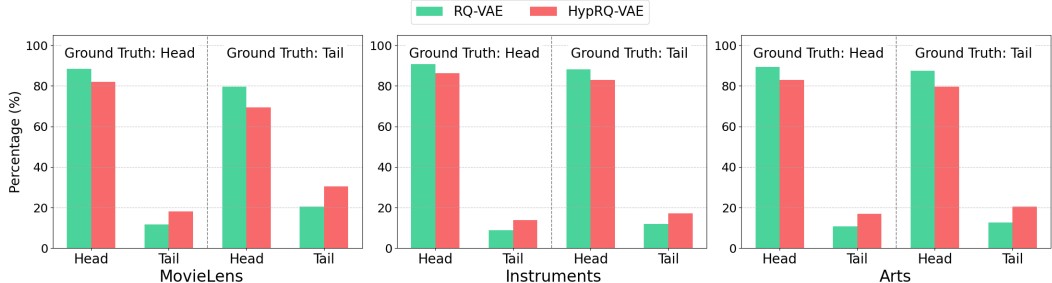

Figure 5: Comparative performance of the RQ-VAE and HypRQ-VAE models regarding top-5 predictions across three datasets, highlighting each model's tendency to recommend popular (head) or niche (tail) items when evaluated against ground truth preferences.

Table 3: Comparison of AQE for H20 and T80 items. The $\Delta_{\text{H-T}}$ column shows the percentage difference between H20 and T80 AQE within each space.

| | # Items | | AQE (Euclidean Space) | | | AQE (Hyperbolic Space) | | | Gain (%) |
|---|---|---|---|---|---|---|---|---|---|
| Dataset | H20 | T80 | H20 | T80 | $\Delta_{\text{H-T}}$ (%) | H20 | T80 | $\Delta_{\text{H-T}}$ (%) | |
| MovieLens | 551 | 2,200 | 0.025,64 | 0.025,68 | −0.16 | 0.001,09 | 0.001,04 | 4.81 | 4.97 |
| Instruments | 1,985 | 7,937 | 0.022,91 | 0.023,67 | −3.32 | 0.001,92 | 0.001,93 | −0.52 | 2.80 |
| Arts | 4,192 | 16,764 | 0.021,78 | 0.022,76 | −4.50 | 0.001,33 | 0.001,34 | −0.75 | 3.75 |

accommodate this sparsity, allowing clustering centers to meaningfully represent both dense popular items and sparse niche items by preserving their structural relationships and maintaining appropriate distance during the quantization process. This geometry naturally organizes items into an implicit hierarchy. Popular items liked by many (e.g., flowers and fruit) are positioned near the hyperbolic origin—the "trunk" of the tree. Conversely, niche items appreciated by specific groups—like a paintbrush for a painter or a guitar for a musician—are placed further out towards the boundary, resembling the "branches" (Yang et al., 2022).

To test this hypothesis, we introduce the Quantization Error (QE) to measure the distance between an item's original embedding and its assigned cluster center (codebook vector). We calculate the Average Quantization Error (AQE) for both H20 and T80 in both spaces. Detailed results are shown in Table 3. While the absolute AQE values are not directly comparable due to their distinct geometric properties, the relative difference within each space is highly informative. The results reveal distinct patterns in how head and tail items are represented across different spaces. In Euclidean space, head items consistently have a lower AQE compared to tail items, confirming they are better represented. However, the hyperbolic space demonstrates a more balanced representation: the disparity between head and tail items is substantially reduced in the Instruments and Arts datasets. Most notably, on MovieLens, tail items achieve a lower AQE than head items, aligning with the massive performance boost observed for tail items in Table 2. This provides strong quantitative evidence that hyperbolic geometry leads to a more balanced and effective representation for both popular and niche items.

## 4.4 ABLATION STUDY

**Length of Semantic IDs.** We conducted an ablation study by varying the length $L$ of the semantic IDs from 3 to 8 (see Figure 6). We can notice that performance rises steadily as $L$ increases from 3 to 6, indicating that shorter identifiers may fail to encode sufficient fine-grained details, thus limiting their expressiveness. However, further increasing $L$ from 6 to 8 results in a performance decline. This is a plausible outcome of error

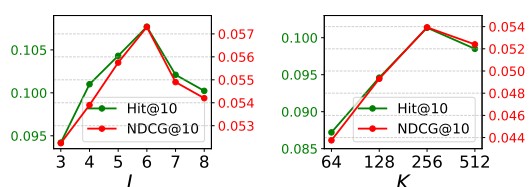

Figure 6: Performance of HYPRQ-VAE over different hyperparameters on MovieLens.

accumulation inherent in autoregressive generation, as the task of accurately generating a longer sequence of codes is inherently more challenging than generating a shorter one, thus affecting the overall item prediction accuracy.

**Codebook Dimension.** We explored the effect of codebook dimension $K$ on the performance of HYPRQ-VAE by testing values of 64, 128, 256, and 512, as shown in Figure 6. The results suggest that a gradual increase in $K$ generally leads to improved performance. The lower performance observed with smaller codebooks may be attributed to insufficient code diversity, limiting the model's capacity to differentiate items effectively. However, excessively large codebooks can lead to a decline in performance, potentially due to increased sensitivity to noise in the item's semantic information, which can result in overfitting on less meaningful aspects.

## 5    RELATED WORK

**Generative Recommendation.** LLMs have shown promising prospects for generative recommendation, where effective item tokenization is crucial. Existing item tokenization research primarily investigates three types of identifiers for item indexing: ID identifiers (Hua et al., 2023), textual identifiers (Tan et al., 2024), and codebook-based identifiers (Rajput et al., 2023). While randomly assigned IDs struggle to encode semantic information and collaborative patterns of items effectively, ID identifiers such as SemID (Hua et al., 2023) and CID (Hua et al., 2023) utilize item semantic information and collaborative signals to construct tree-like structures for generating identifiers. However, these fixed, unlearnable structures struggle to efficiently represent item similarity and adapt to new items. Codebook-based identifiers leverage codebooks and attempt to integrate semantic information and collaborative signals during the training process. However, they overlook the misalignment introduced by incorporating collaborative signals into fixed code sequences, as well as the code assignment bias. For example, TIGER (Rajput et al., 2023) employs standard RQ-VAE to generate the identifiers of items, LC-Rec (Zheng et al., 2024) uses the Sinkhorn-Knopp Algorithm to employ a uniform distribution constraint in the last layer to mitigate item indexing conflicts, and LETTER (Wang et al., 2024) introduces regularizers to enhance code assignment diversity. However, these methods typically operate in Euclidean space. In this work, we propose a hyperbolic model that generates semantic tokenizations for items with hyperbolic representation.

**Hyperbolic Learning.** Growing evidence shows that many real-world datasets exhibit non-Euclidean structure and benefit from hyperbolic representations (Ganea et al., 2018; Shimizu et al., 2020). Early work applied Poincaré embeddings to capture hierarchical relationships in complex networks (Nickel & Kiela, 2017; Tifrea et al., 2018; Balazevic et al., 2019). More recently, hyperbolic geometry has been incorporated into advanced deep architectures, such as the recurrent neural network (He et al., 2024), attention network (Gulcehre et al., 2018), and graph neural network (Chami et al., 2019) to exploit its ability to model hierarchy and hierarchy-like structure. In recommender systems, hyperbolic approaches such as HGCF (Sun et al., 2021) and HICF (Yang et al., 2022) have demonstrated superior preference modeling over Euclidean baselines. In contrast to these studies, which focus on traditional collaborative filtering, our work is the first to integrate hyperbolic embeddings into the generation of semantic IDs, opening new avenues for hyperbolic-driven generative recommendation.

## 6    CONCLUSION

In this paper, we proposed HYPRQ-VAE, the first framework to integrate hyperbolic geometry into residual quantized variational autoencoders for item indexing in generative recommender systems. By leveraging the exponentially expanding capacity of hyperbolic space, our model effectively addresses the semantic misalignment between language models and recommendation tasks, while simultaneously improving performance for long-tail items. Comprehensive experiments across multiple benchmark datasets validate our approach HYPRQ-VAE consistently outperforms Euclidean-based models, particularly in tail-item recommendation. Our analysis confirms this is due to a more balanced and effective quantization of both popular and niche items in hyperbolic space. This work highlights the potential of hyperbolic geometries to enhance generative recommendation and opens new avenues for developing more expressive and equitable item indexing methods.

## ETHICS STATEMENT

Our study focuses on developing and evaluating new methods for recommender systems, with experiments conducted on publicly available benchmark datasets. No human subjects, personally identifiable information, or sensitive user data were involved. We acknowledge that recommendation technologies can potentially amplify biases or reinforce popularity imbalances; our work explicitly addresses this concern by investigating methods to better model long-tail items, thereby promoting fairness and diversity in recommendations. We have taken care to ensure compliance with ethical standards in dataset usage, legal requirements, and research integrity, and we disclose no conflicts of interest or external sponsorship that could create undue influence.

## REPRODUCIBILITY STATEMENT

We have made significant efforts to ensure the reproducibility of our work. The full source code and processed data are released through an anonymous GitHub repository, enabling direct replication of our experiments. Detailed implementation settings, including model architecture, training procedures, are provided in Section 4, with additional implementation details and hyperparameters in Appendix C. The datasets used are publicly available. Together, these resources allow independent researchers to reproduce our results and validate our findings.

## THE USE OF LARGE LANGUAGE MODELS (LLMS)

LLMs were employed exclusively as general-purpose writing aids, assisting with grammar correction, and improving readability. They were not used for research ideation, methodology design, data analysis, or interpretation of results. All conceptual contributions, technical methods, experiments, and findings were entirely authored and verified by the research team.

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

## A DATASET STATISTICS

Table 4: Statistics of three real-world benchmarks.

| Datasets | # Users | # Items | | | # Interactions | # Density |
|---|---|---|---|---|---|---|
| | | All | H20(%) | T80(%) | | |
| Movie Lens | 6,040 | 2,751 | 59.97 | 40.03 | 1,000,000 | 6.018% |
| Instruments | 24,773 | 9,922 | 61.00 | 39.00 | 206,153 | 0.084% |
| Arts | 45,142 | 20,956 | 57.65 | 42.35 | 390,832 | 0.041% |

We evaluate our model on three widely adopted real-world recommendation datasets, including Instruments, Arts from the Amazon[4] dataset (Ni et al., 2019), and MovieLens. Each item includes both a title and a description; for MovieLens, we augment each movie with its plot overview from TMDB[5] as the description of it. Following previous work (Hou et al., 2022), we filter out users and items with less than five interactions, then order each user's remaining interactions chronologically to form behavior sequences. To align with baseline settings, we truncate all sequences to a maximum length of 20. The statistics of the dataset are in Table 4, where H20 and T80 denote the average ratio of the head items and tail items appearing in user's preference.

## B BASELINES

We compare our method with the following baseline methods, including standard sequential recommenders and LLM-based models:

- **MF** (Rendle et al., 2012) decomposes the user-item interactions into the user embeddings and the item embeddings in the latent space.
- **Caser** (Tang & Wang, 2018) employs convolutional neural networks to capture users' spatial and positional information.
- **SASRec** (Kang & McAuley, 2018) employs self-attention mechanisms to capture long-term dependencies in user interaction history.
- **P5-TID** (Hua et al., 2023) uses the title of items as textual identifiers for the LLM-based generative recommender model.
- **P5-CID** (Hua et al., 2023) incorporates collaborative signals into the identifier for LLM-based generative recommender models through a spectral clustering tree derived from item co-appearance graphs.
- **TIGER** (Rajput et al., 2023) introduces codebook-based identifiers via RQ-VAE, which quantizes semantic information into code sequence for LLM-based generative recommendation.
- **LC-Rec** (Zheng et al., 2024) uses codebook-based identifiers and auxiliary alignment tasks to better utilize knowledge in LLMs by connecting generated code sequences with natural language.
- **LETTER** (Wang et al., 2024) integrates hierarchical semantics, collaborative signals, and code assignment diversity to conduct effective item tokenization for LLM-based generative recommendation.

## C EXPERIEMENT DETAILS

**Parameter Settings.** We use the default hyperparameter settings in the released code of the corresponding publications. This ensures that our experiments remain consistent and comparable. For the proposed method, we have listed the hyperparameter settings in the Table 5 below for Stage 1 (Generation of Semantic IDs) and Stage 2 (Generative Recommendation with Semantic IDs).

---

[4]https://jmcauley.ucsd.edu/data/amazon/
[5]https://www.themoviedb.org/

Table 5: Training setup across two stages.

| Phase | Component | Value |
|-------|-----------|-------|
| Stage 1 | Input | Title + Description |
| | $\beta$ | 0.25 |
| | $\gamma$ | 4 |
| | Layer of codebooks $L$ | 4 |
| | Dimension of each codebook $K$ | 256 |
| | Optimizer | Adam |
| | LR | 1e−3 |
| | Epoch | 5000 |
| Stage 2 | Model | LLaMA2-7B + LoRA |
| | LoRA Config | $r = 8, \; \alpha = 32, \; \text{dropout} = 0.05$ |
| | Optimizer | AdamW |
| | LR | 1e−4 (cosine schedule) |
| | Epochs | 4 |

**Prompt Template.** Our approach is framed in an LLM-based generative manner, where sequential item prediction serves as the primary tuning objective. Each instruction is personalized by combining a user's chronological interaction history, represented as a sequence of the semantic IDs we generate, with a natural language prompt. To enhance the model's generalizability and robustness, we designed a diverse set of prompt templates. An example of one such template is provided below:

> **Prompt Template**
>
> **Instruction:** The user has interacted with the following items in sequence: {semantic IDs of interaction items}. Based on this interaction history, please predict the next item the user is most likely to engage with.
>
> **Response:** {semantic ID of target item}

Table 6: Performance Comparison of RQ-VAE and HypRQ-VAE on Three Datasets (Top-5 Predictions).

| Dataset | Method | $GT_H$-$R_H$ | $GT_H$-$R_T$ | $GT_T$-$R_H$ | $GT_T$-$R_T$ |
|---------|--------|--------------|--------------|--------------|--------------|
| Movie-Lens | RQ-VAE | 88.32% | 11.67% | 79.48% | 20.51% |
| | HypRQ-VAE | 81.88% | 18.09% | 69.43% | 30.57% |
| Instruments | RQ-VAE | 91.21% | 8.79% | 88.15% | 11.85% |
| | HypRQ-VAE | 86.18% | 13.82% | 82.82% | 17.18% |
| Arts | RQ-VAE | 89.29% | 10.71% | 87.41% | 12.59% |
| | HypRQ-VAE | 83.03% | 16.97% | 79.53% | 20.47% |

## D  PERFORMANCE ANALYSIS ON HEAD VS. TAIL ITEMS

We calculated the proportion of head and tail items within the top-$k$ predictions. The results are presented in the tables below (Table 6 and Table 7), where $GT_H$ indicates that the ground truth (target item) is a head item, and $GT_T$ indicates a tail item. $R_H$ and $R_T$ refer to the ratio of H20 and T80 items in the top-$k$ predicted list.

The results consistently show that the HypRQ-VAE model predicts a higher proportion of tail items compared to the standard RQ-VAE. This is evident across all three datasets and for both top-5 and

Table 7: Performance Comparison of RQ-VAE and HypRQ-VAE on Three Datasets (Top-10 Predictions).

| Dataset | Method | $GT_H$-$R_H$ | $GT_H$-$R_T$ | $GT_T$-$R_H$ | $GT_T$-$R_T$ |
|---|---|---|---|---|---|
| Movie-Lens | RQ-VAE | 86.26% | 13.73% | 77.82% | 22.17% |
| | HypRQ-VAE | 79.32% | 20.65% | 67.23% | 32.75% |
| Instruments | RQ-VAE | 88.65% | 11.35% | 85.24% | 14.76% |
| | HypRQ-VAE | 83.88% | 16.12% | 80.14% | 19.86% |
| Arts | RQ-VAE | 86.34% | 13.66% | 84.39% | 15.61% |
| | HypRQ-VAE | 79.96% | 20.04% | 76.68% | 23.32% |

top-10 predictions, regardless of whether the ground truth item is a head or tail item. When the ground truth is a head item ($GT_H$), RQ-VAE's predictions are heavily skewed towards other head items, exhibiting a high $R_H$. In contrast, HypRQ-VAE demonstrates a lower $R_H$ and a higher $R_T$ in its predictions compared to RQ-VAE. This indicates that HypRQ-VAE is more effective at incorporating a mix of tail items. When the ground truth is a tail item ($GT_T$), the difference is even more pronounced. HypRQ-VAE significantly increases the proportion of tail items in its predictions, with its $R_T$ values being substantially higher than those of RQ-VAE. This demonstrates HypRQ-VAE's enhanced ability to recommend less popular, long-tail items, which is particularly beneficial for long-tail recommendation scenarios.

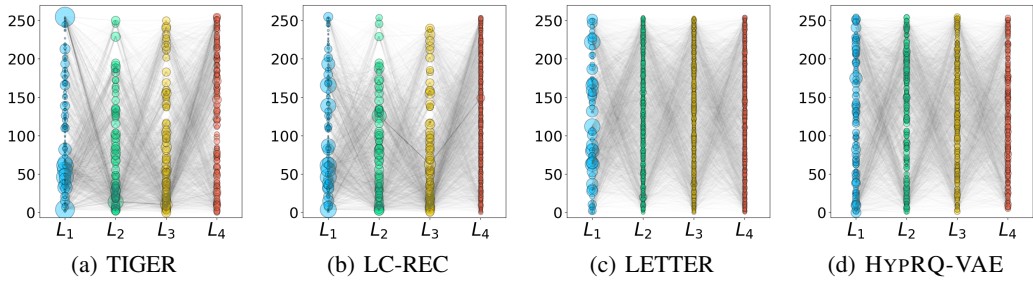

| (a) TIGER | (b) LC-REC | (c) LETTER | (d) HYPRQ-VAE |

Figure 7: Code paths across codebook layers for various methods on MovieLens. Compared with RQ-VAE, HYPRQ-VAE achieves a remarkably even token distribution without the need for additional regularizers or constraints.

## E VISUALIZATION OF CODEBOOK UTILIZATION

We further visualize the code path across codebook layers for different methods. The number of nodes for each layer matches the size of the corresponding codebook, and the node size reflects the degree of the respective token. The results are shown in Figure 7, and we can notice that: When compared to TIGER, LC-Rec shows a more uniformly distributed token assignment in its last layer, and LETTER achieves a more balanced distribution across its last three layers. Notably, HYPRQ-VAE as a hyperbolic adaptation of RQ-VAE, achieves a remarkably even token distribution without the need for additional regularizers or constraints.

Moreover, we investigated different initialization strategies and discovered an interesting phenomenon: when employing uniform distribution initialization, the code identifiers for each layer exhibited a distinctive distribution pattern (as shown in Figure 8). Furthermore, as the number of layers increased, additional representational space became

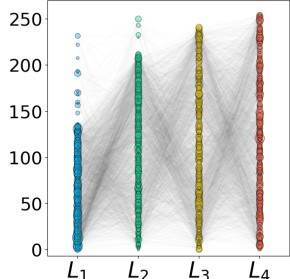

Figure 8: Uniform Distribution Initialization.

necessary to differentiate increasingly granular levels of information. This observation suggests a promising direction for future work, specifically, the adaptive adjustment of dimensional capacity at each layer to optimize spatial utilization throughout the network architecture. Specifically, the previous layers might focus on broad semantic categories, requiring less space, while subsequent layers refine these into finer levels of information, demanding increased dimensionality. An adaptive approach could dynamically allocate resources, potentially leading to more compact and efficient representations.

