# OpenReview forum: "HypRQ-VAE: Long-Tail-Aware Item Indexing for Generative Recommender Systems"
_ICLR.cc/2026/Conference — ICLR 2026 Conference Withdrawn Submission_

### Official Review · Reviewer_WA8E · 2025-10-23

**Soundness:** 2
**Presentation:** 2
**Contribution:** 2
**Rating:** 2
**Confidence:** 4

**Summary:**

This paper introduces a novel item indexing method based on hyperbolic geometry to enhance generative recommendation systems. By extending the traditional RQ-VAE model from Euclidean to hyperbolic space, the proposed model, HypRQ-VAE, effectively addresses the challenge of long-tail item indexing. Experimental results on three real-world datasets show that HypRQ-VAE outperforms several ID-based and generative recommendation models.

**Strengths:**

1. Clear motivation: The paper explores the potential of hyperbolic space for addressing long-tail item indexing, a novel direction that could significantly improve item representation in generative recommendation systems.

2. Well written and well structured: The paper is well-organized, with a clear introduction that outlines the limitations of prior work and the motivation for the proposed approach. The background section is effective in preparing readers for the methodology, which is presented clearly and logically. The experimental section is also easy to follow.

3. Promising experimental results: The proposed HypRQ-VAE model achieves stronger recommendation accuracy compared to established baselines on real-world datasets. The paper includes ablation studies that help to understand the impact of key design choices and hyperparameters on model performance.

4. Reproducibility: The authors provide code during the review phase, which supports reproducibility and transparency in their findings.

**Weaknesses:**

1. Weak Contribution and Justification for Hyperbolic Space: Although the proposed model shows improved performance, its contribution seems limited to a straightforward extension of RQ-VAE into hyperbolic space. The paper lacks sufficient justification for why hyperbolic geometry is particularly suited to address the long-tail item problem. More discussion is needed on the advantages of hyperbolic space over Euclidean space and how this choice contributes uniquely to the model's performance. This would strengthen the rationale behind the proposed approach and differentiate it from prior work.

2. Lack of Comparison with Relevant Baselines: The paper does not compare its method against important recent work, such as Liu et al. (2024) (Llmesr: Large language models enhancement for long-tailed sequential recommendation. NeurIPS 2024), which leverages large language models (LLMs) to address long-tail item issues in sequential recommendations. Including this comparison would provide a clearer perspective on the relative strengths of the proposed model.

3. No Statistical Significance Testing: Table 1 reports performance gains, but there is no statistical significance testing to validate these results. Given that some of the performance improvements are small, it’s crucial to conduct significance tests and clarify the number of runs performed to ensure that the reported gains are not due to random variations or seed biases.

4. Unclear Evaluation on Long-Tail Items: The evaluation of long-tail item performance lacks clarity and is insufficiently convincing. For example, why is TIGER used as the baseline in Table 2 when LETTER appears to be the stronger baseline in Table 1? Additionally, while the proposed method is designed to improve long-tail item performance, the actual improvements are modest in many cases, calling into question the method’s effectiveness for this specific problem. Further analysis is needed to support the claim that the proposed method provides substantial benefits for long-tail items.

**Questions:**

Please see the review.

---

### Official Review · Reviewer_naXt · 2025-10-31

**Soundness:** 2
**Presentation:** 1
**Contribution:** 2
**Rating:** 4
**Confidence:** 3

**Summary:**

This paper presents HYPRQ-VAE, a framework for generative recommendation that addresses the semantic mismatch between language models and discrete item indices. Unlike prior approaches that embed items in Euclidean space, the authors propose learning item representations in hyperbolic space to better capture the power-law distribution and hierarchical structure of real-world item catalogs. The model leverages hyperbolic geometry and residual quantization within a variational autoencoder framework to preserve both semantic richness and long-tail item fidelity. Experimental results on three benchmark datasets show improvements in recommendation accuracy.

**Strengths:**

The paper identifies a limitation in current LLM-based recommender systems and proposes a strategy using hyperbolic geometry. The integration of hyperbolic space with residual quantization in a VAE framework is original and well-motivated by the structure of user-item interactions. The proposed method shows empirical gains in recommending long-tail items.

**Weaknesses:**

1. While the paper proposes to embed item IDs in hyperbolic space, this appears to be the primary contribution. The use of hyperbolic geometry in recommendation systems has been well explored in prior work (e.g., [1][2]), and the combination with long-tail modeling or language models is also not new. As such, the contribution may be considered incremental.

2. The method is evaluated on only three benchmark datasets, which may not sufficiently show its generalizability across diverse recommendation scenarios.

[1] Cheng W, Qin Z, Wu Z, et al. Large language models enhanced hyperbolic space recommender systems[C].Proceedings of the 48th International ACM SIGIR Conference on Research and Development in Information Retrieval. 2025: 1944-1953.

[2] Yang M, Feng A, Xiong B, et al. Enhancing llm complex reasoning capability through hyperbolic geometry[C].ICML 2024 Workshop on LLMs and Cognition. 2024.

**Questions:**

1. The token reassignment strategy for handling collisions involves a cascading, multi-level backtracking process. How sensitive is the model performance to this procedure? Is it stable under varying numbers of colliding items?

2. The experimental evaluation is conducted on only three datasets, two of which are from the same domain (Amazon). This raises concerns about the diversity of the evaluation. The authors should include datasets from other domains (e.g., news, music, or social recommendations) to better show its generalizability.

3. The authors should include a comparison with the method proposed in [1], which is clearly relevant to the problem setting addressed in this work.

---

### Official Review · Reviewer_Nqtc · 2025-10-31

**Soundness:** 2
**Presentation:** 1
**Contribution:** 1
**Rating:** 2
**Confidence:** 4

**Summary:**

This paper investigates semantic ID generation in generative recommendation. The authors identify the issue of long-tailed semantic ID distributions in existing RQ-VAE–based methods, where they suffer from optimization issue in Euclidean space. To mitigate this problem, the authors take the initial attempt to leverage hyperbolic space for semantic ID generation, along with a two-stage training strategy to enhance semantic integration. Extensive experiments on three real-world datasets demonstrate (1) the effectiveness of hyperbolic models in generating semantic IDs and (2) the superiority of hyperbolic space over Euclidean space, particularly for long-tailed items in recommendation tasks.

**Strengths:**

1. The paper tackles a fundamental problem in generative recommendation, i.e., semantic ID generation, which is an important and impactful research direction.

2. The authors present an initial exploration of applying hyperbolic models to semantic ID generation, offering new insights into how hyperbolic space can better capture item relationships.

3. The authors conduct comprehensive experiments that validate the effectiveness of hyperbolic RQ-VAE in learning semantic IDs and demonstrate improved performance on long-tailed item recommendations.

**Weaknesses:**

1. The motivation is vague and lacks detailed analysis of how Euclidean space specifically affects optimization. The precise research problem being solved is therefore unclear.

2. The method design appears rather intuitive and incremental, as it can be easily derived from prior works that already apply hyperbolic space for optimization in recommendation [1].

3. Several key descriptions related to the motivation and methodology are confusing or underspecified (see Questions section for details).

4. The two-stage training strategy is questionable, since the second phase still relies on Euclidean optimization, which may reintroduce the issues the paper aims to solve.

[1] Hyperbolic Neural Collaborative Recommender. TKDE 2022

**Questions:**

1. What are the practical consequences of having a long-tailed distribution in semantic IDs? How does it affect the downstream recommendation performance?

2. Figure 1 is a bit confusing. How is the frequency of “semantic” computed? Do “semantic” and “popularity” share the same horizontal axis?

3. A more detailed explanation of why Euclidean space struggles with long-tail distributions is essential for readers to fully grasp the motivation and the rationale for adopting hyperbolic space.

4. The paper should elaborate on the inherent capability of hyperbolic geometry to model hierarchical structures. Currently, this claim is supported only by citations, without clear conceptual integration.

5. The paper suggests that hyperbolic training leads to more uniform semantic ID distributions, but neither provides a theoretical justification nor empirical evidence to support this. Even in Figure 7, the HYPRQ-VAE code distribution across L2–L4 appears less uniform than that of LETTER.

6. The second-stage recommender training remains problematic. Since token embeddings are still optimized in Euclidean space, popularity bias may persist or even be amplified, leading to a misalignment between the semantic ID space (hyperbolic) and the recommender model space (Euclidean). This critical issue should be discussed and addressed more explicitly.

---

### Official Review · Reviewer_d2xh · 2025-11-01

**Soundness:** 3
**Presentation:** 3
**Contribution:** 2
**Rating:** 4
**Confidence:** 3

**Summary:**

This paper proposes a novel hyperbolic residual-quantized VAE for generative recommendation, which can handle the long-tail items better in learning semantic IDs. Specifically, the proposed method map item embeddings into hyperbolic space, where exponential volume expansion better models the hierarchical and power-law structure of user-item interactions. By performing residual quantization within this space, HYPRQ-VAE generates discrete semantic IDs that capture both head and tail items effectively. Comparison with Tiger, an RQ-VAE model in Euclidean space, validates its effectiveness.

**Strengths:**

1. Novel geometric formulation. The employment of hyperbolic geometry in item tokenization is novel.

2. Motivation. The paper clearly demonstrates how hyperbolic space’s exponential expansion and hierarchical topology align with the power-law distribution of item popularity, making the proposed a conceptually elegant and theoretically grounded approach to handle the long-tail problem in generative recommendation.

**Weaknesses:**

1. Challenge. The proposed method replaces the Euclidean space with the Hyperbolic space in code lookup. However, such employment seems straightforward, and there is no much technical challenge.

2. Analysis. Even though it's well-known that the Hyperbolic space has better properties in handling long-tail items, concrete examples or intuitive explanations on the advantage of the proposed method v.s. the Tiger would be great.

3. Evaluation. The experiments are limited, with only one baseline model.

**Questions:**

check the analysis part of the Weakness.

---

### Note · Authors · 2025-12-07

**Comment:**

We would like to withdraw our manuscript from consideration at ICLR.

**Withdrawal Confirmation:**

I have read and agree with the venue's withdrawal policy on behalf of myself and my co-authors.